# Physiological Basis of Extracorporeal Membrane Oxygenation and Extracorporeal Carbon Dioxide Removal in Respiratory Failure

**DOI:** 10.3390/membranes11030225

**Published:** 2021-03-22

**Authors:** Barbara Ficial, Francesco Vasques, Joe Zhang, Stephen Whebell, Michael Slattery, Tomas Lamas, Kathleen Daly, Nicola Agnew, Luigi Camporota

**Affiliations:** 1Department of Adult Critical Care, Guy’s and St. Thomas’ NHS Foundation Trust, King’s Health Partners, London SE1 7EH, UK; barbara.ficial@gmail.com (B.F.); francesco.vasques@hotmail.it (F.V.); joe.zhang@gstt.nhs.uk (J.Z.); stephen.whebell@gstt.nhs.uk (S.W.); michael.slattery@gstt.nhs.uk (M.S.); kathleen.daly@gstt.nhs.uk (K.D.); nicola.agnew@gstt.nhs.uk (N.A.); 2Department of Critical Care, Unidade de Cuidados Intensivos Polivalente, Egas Moniz Hospital, Rua da Junqueira 126, 1300-019 Lisbon, Portugal; tomaslamas2009@gmail.com; 3Division of Centre of Human Applied Physiological Sciences, King’s College London, London SE1 7EH, UK

**Keywords:** extracorporeal life support (ECLS), extracorporeal membrane oxygenation (ECMO), extracorporeal CO_2_ removal (ECCO_2_R), acute respiratory distress syndrome (ARDS), chronic obstructive pulmonary disease (COPD)

## Abstract

Extracorporeal life support (ECLS) for severe respiratory failure has seen an exponential growth in recent years. Extracorporeal membrane oxygenation (ECMO) and extracorporeal CO_2_ removal (ECCO_2_R) represent two modalities that can provide full or partial support of the native lung function, when mechanical ventilation is either unable to achieve sufficient gas exchange to meet metabolic demands, or when its intensity is considered injurious. While the use of ECMO has defined indications in clinical practice, ECCO_2_R remains a promising technique, whose safety and efficacy are still being investigated. Understanding the physiological principles of gas exchange during respiratory ECLS and the interactions with native gas exchange and haemodynamics are essential for the safe applications of these techniques in clinical practice. In this review, we will present the physiological basis of gas exchange in ECMO and ECCO_2_R, and the implications of their interaction with native lung function. We will also discuss the rationale for their use in clinical practice, their current advances, and future directions.

## 1. Introduction

Respiratory failure is a condition in which the respiratory system is unable to maintain adequate gas exchange to satisfy the body’s metabolic demands. Of patients with respiratory failure, over 33% receive mechanical ventilation for more than 12 h and a significant proportion (10%) develop Acute Respiratory Distress Syndrome (ARDS) with a mortality of 35–50% [1]. Furthermore, in patients with very large shunt fraction and dead space, hypoxaemia or hypercapnia may be refractory to changes in inspiratory oxygen and high “intensity” mechanical ventilation (i.e., high airway pressures, respiratory rate, mechanical power) [2,3]. In addition, those who develop more severe or life-threatening respiratory failure cannot always be managed safely with conventional mechanical ventilation. In these patients, the use of extracorporeal support via a “membrane lung”—extracorporeal membrane oxygenation (ECMO)—permits to achieve sufficient oxygen delivery and carbon dioxide removal. At the same time, it allows a reduction in the intensity of ventilation, thus diminishing the risk of ventilator-induced lung injury (VILI) and allowing native lung recovery.

The use of ECMO, the number of ECMO centres and the use of mobile ECMO has increased exponentially since 2009 [4] following the H_1_N_1_ influenza A pandemic [5,6] and the publication of the CESAR trial [7]. Peek at al showed greater survival without disability (63% vs. 47%; relative risk 0.69) and greater quality-adjusted life-years (QALYs) at 6 month when ECMO was offered. Several non-randomised, but matched cohort studies showed safety and efficacy of ECMO [5,6,8]. More recently, the largest randomized trial on ECMO (EOLIA Trial) [9] included 249 patients and showed a clinically large—although not statistically significant—reduction in mortality with ECMO (35% vs. 46%; RR, 0.76). A subsequent post-hoc Bayesian analysis of the EOLIA trial showed that early VV-ECMO provides mortality benefit in patients with very severe ARDS as per EOLIA inclusion criteria [10]. This benefit has been further confirmed by a recent meta-analysis which reported a significantly lower 60-day mortality in patients who received ECMO (RR, 0.73 [95% CI, 0.58–0.92]; *p* = 0.008) [11].

ECMO is a supportive treatment and can represent a “bridge” either to healing of natural organs (bridge-to-recovery), to long-term devices (bridge-to-destination) or to organ replacement (bridge-to-transplantation). Occasionally, when the aetiology is unclear, it becomes a bridge to make a decision by allowing initial stabilisation of the patient and subsequent re-evaluation. Eventually, for some patients, ECMO becomes a bridge to palliation when no further therapeutic options are available.

## 2. Definitions

Extracorporeal membrane oxygenation (ECMO) and extracorporeal carbon dioxide removal (ECCO_2_R) are two modalities of extracorporeal life support (ECLS) which can partially (ECCO_2_R) or fully (ECMO) substitute the gas exchange function of the native lung (Figure 1) [12].


(1)*In veno-venous extracorporeal membrane oxygenation (VV-ECMO),* deoxygenated blood is drained from a central vein (e.g., inferior or superior vena cava), pumped through a “membrane lung” or oxygenator, and then reinfused—fully oxygenated and decarboxylated—into a central vein. The gas exchange is driven by the diffusion gradient across the membrane. Fresh gas (sweep gas flow, SGF) flows, countercurrent to the blood, in the lumen of the membrane’s hollow fibers, therefore maintaining a favourable gradient for oxygenation and decarboxylation. To achieve this, the blood flows used in ECMO range between 3 and 7 L/min.(2)*Extracorporeal carbon dioxide removal (ECCO_2_R)* operates in a similar way to VV-ECMO but at much lower blood flow (usually < 1–1.5 L/min). The main difference between VV-ECMO and ECCO_2_R consists in the fact that CO_2_ removal has a linear kinetic and is more efficient than oxygenation: for this reason, ECCO_2_R can be delivered at much lower flows and with smaller cannulae. Although ECCO_2_R does not provide significant oxygenation, it is able to remove 25–50% of the metabolically produced carbon dioxide.(3)*Veno-arterial extracorporeal membrane oxygenation (VA-ECMO*): VA-ECMO provides support for the heart (e.g., biventricular failure). Hybrid configurations for peripheral ECMO (V-VA-ECMO), where there is drainage of the venous system and return into a central artery and a central vein to assist both heart and lung (e.g., severe ARDS and septic cardiomyopathy with cardiogenic shock).


From this point forward when we mention ECMO we intend the veno-venous (VV) configuration.

## 3. Physiology of ECMO

### 3.1. Oxygenation

#### 3.1.1. Oxygen Delivery and Demand

Oxygen demand in critically ill patients can vary between 200 and 450 mL/min depending on the underlying disease and metabolic state. Therefore, to maintain an oxygen delivery to oxygen demand ratio (DO_2_/VO_2_ ratio) of ~3, the amount of oxygen delivered to the tissues must be 600–1350 mL/min. The total DO_2_ is equal to:(1)$${\mathrm{DO}}_{2}={\mathrm{CaO}}_{2}\times \mathrm{CO}$$

In patients with ARDS not on VV-ECMO, the CaO_2_ depends on the shunt fraction of the native lung, and on the oxygen content of the mixed venous blood:(2)$${\mathrm{CaO}}_{2}=\left[{\mathrm{CcO}}_{2}\times \left(\mathrm{Qt}-\mathrm{Qs}\right)\right]+\left({\mathrm{CvmixO}}_{2}\times \mathrm{Qs}\right)$$
where Qt−Qs is the portion of the cardiac output going through the ventilated lung parenchyma, and Qs is the portion of the cardiac output shunting through non ventilated lung areas. Oxygen consumption (VO_2_) can be estimated as:(3)$${\mathrm{VO}}_{2}=\mathrm{CO}\times \left({\mathrm{CaO}}_{2}-{\mathrm{CvmixO}}_{2}\right)$$

In patients fully dependent on ECMO (i.e., with no residual native lung function) the CaO_2_—in its simplest form, without accounting for recirculation (see below)—is:(4)$${\mathrm{CaO}}_{2}=C_{\mathrm{post}-\mathrm{oxy}}O_{2}\times \mathrm{ECBF}+{\mathrm{CvO}}_{2}\times \left(\mathrm{CO}-\mathrm{ECBF}\right)$$

This formula is analogous to the formula (2) where ECMO blood flow is noted as ECBF and CO is the cardiac output of the patient, CvO_2_ is the content of oxygen in the venous blood and C_post-oxy_O_2_ content of the blood exiting the oxygenator.

The VO_2_ (mL/min) of the membrane lung (VO_2ML_) is proportional to the ECMO blood flow (ECBF) and to the difference between the oxygen content of the blood entering the oxygenator (C_pre-oxy_O_2_) and the content of the blood exiting the oxygenator (C_post-oxy_O_2_) (Figure 2):(5)$${\mathrm{VO}}_{2\;\mathrm{ML}}=(C_{\mathrm{post}-\mathrm{oxy}}O_{2}-C_{\mathrm{pre}-\mathrm{oxy}}O_{2})\times \mathrm{ECBF}$$

By rearranging Equation (4):(6)$${\mathrm{CaO}}_{2}=\left(\frac{\mathrm{ECBF}}{\mathrm{CO}}\right)\times C_{\mathrm{post}-\mathrm{oxy}}O_{2}+\left[1-\left(\frac{\mathrm{ECBF}}{\mathrm{CO}}\right)\right]\times {\mathrm{CvO}}_{2}$$

Using this formula, the oxygen content is expressed in terms of the ratio between the ECBF and the cardiac output—in a similar way as the shunt equation of the native lung. ECBF represents the equivalent of Qt-Qs and CO-ECBF is the equivalent of Qs, as blood is effectively shunted from the ECMO.

It becomes clear that oxygen content depends not only on the ECMO blood flow and the content of oxygen in the venous blood (i.e., the relationship between oxygen delivery and tissue oxygen extraction), but also on the ratio between ECBF and the patient’s cardiac output. To understand this concept, we have to consider how the venous return, equal to the patient cardiac output, is “split” into two components: (1) one part—equal to the ECBF—will pass through the oxygenator and therefore will return to the right atrium fully saturated with oxygen (S_post-oxy_ = 100%; P_post-oxy_O_2_ ~ 60–70 kPa or 450–525 mm Hg); (2) the second part of the venous return—which is equal to the amount of flow that exceeds the ECBF (i.e, CO − ECBF) will have the saturation of the venous blood. Therefore, the mixed venous blood of the patient (the oxygenation of the blood in the pulmonary artery) will be a mixed “weighed average” of the two in a proportion that will depend on: the ratio between ECBF and CO; the venous oxygenation and the functioning of the membrane (i.e., the ability to fully oxygenate the venous blood).

#### 3.1.2. Shunt of the Membrane Lung

Membrane function is determined by its characteristics but, over time, is also affected by “aging”: clots accumulation and protein binding will occlude some of the hollow fibres, preventing the flow of fresh gas through them. As a result, part of the membrane lung (similarly to the native lung) will be perfused, but not “ventilated”—i.e., not in contact with the sweep gas flow. This phenomenon will generate a “shunt of the membrane lung”, that can be calculated utilising Riley’s model of shunt of the natural lung [13,14,15]:(7)$$\frac{Q_{s}}{\mathrm{ECBF}}=\frac{C_{\mathrm{capillary}-\mathrm{oxy}}O_{2}-C_{\mathrm{post}-\mathrm{oxy}}O_{2}}{C_{\mathrm{capillary}-\mathrm{oxy}}-C_{\mathrm{pre}-\mathrm{oxy}}O_{2}}$$

C_capillary-oxy_O_2_ is the oxygen content of the “capillary” blood (in the oxygenator) that is in equilibrium with the “alveolar” gas (in the hollow fibers). It can be calculated from the “alveolar” oxygen partial pressure:(8)$$P_{\mathrm{capillary}-\mathrm{oxy}}O_{2}=\left({\mathrm{FiO}}_{2}\times P_{\mathrm{barometric}-H_{2}O}\right)-(P_{\mathrm{post}-\mathrm{oxy}}{\mathrm{CO}}_{2}\times {\mathrm{VO}}_{2}/{\mathrm{VCO}}_{2})$$

Changes in the membrane lung performance can be detected by monitoring P_post-oxy_O_2_, VO_2ML_ and the Qs/ECBF over time.

#### 3.1.3. Recirculation

If we go one step further: the amount of oxygen entering the membrane lung ($\mathrm{ECBF}\times C_{\mathrm{pre}-\mathrm{oxy}}O_{2}$) derives from two sources. The first one is represented by the systemic venous blood from the tissues $\left(\mathrm{ECBF}-\mathrm{Qr}\right)\times {\mathrm{CvO}}_{2}$. The second one is constituted by recirculating blood flow (${\;Q}_{r}\times (C_{\mathrm{post}-\mathrm{oxy}}O_{2})$): a variable portion of the fully oxygenated blood ejected by the return cannula is immediately re-aspirated by the drainage cannula; this is due to physical proximity of the two cannulae. Recirculation is defined as the fraction of the ECBF that derives from the blood ejected from the return cannula—which is already oxygenated and decarboxylated by the first passage through the membrane lung. Recirculation raises the oxygen content of the blood entering the membrane lung and therefore reduces the efficiency of the gas transfer; firstly, because the oxygen gradient across the membrane will be reduced, and secondly because the effective ECBF (i.e., the ECBF that reaches the right atrium) is no longer the entirety of the ECBF but ECBF −Qr. These two effects will jointly reduce the VO_2ML_ (Equation (5)).

Clinically significant recirculation should be suspected when, despite sweep gas flow F_d_O_2_ = 1:Low SaO_2_ and poor increase in SaO_2_ at higher ECBFIncrease in S_pre-oxy_O_2_ and reduction in SaO_2_ over timeP_pre-oxy_O_2_ > 10% of P_post-oxy_O_2_S_pre-oxy_O_2_ > 75% with SaO_2_ < 85%

#### 3.1.4. Gas Exchange in Partially Dependent Patients

As already discussed, the oxygen reaching the right atrium is the result of the admixture of blood coming from the membrane lung and blood coming from the tissues. If we also account for recirculation, we will have:(9)$${\mathrm{CvmixO}}_{2}\times \mathrm{Qt}=[{\mathrm{CvO}}_{2}\left(\mathrm{Qt}-\mathrm{ECBF}+\mathrm{Qr}\right]+C_{\mathrm{post}-\mathrm{oxy}}O_{2}\times \left(\mathrm{ECBF}-\mathrm{Qr}\right)$$

In patients partially dependent on ECMO, the blood flow distributes once again through the pulmonary arteries. A portion perfuses the healthy lung, capable of gas exchange, and it equilibrates with the PO_2_ of the open ventilated alveoli (${\mathrm{CcO}}_{2}\times \left(\mathrm{Qt}-\mathrm{Qs}\right)$). However, part of the blood flow will perfuse the diseased non-ventilated (shunted) lung areas (${\mathrm{CvmixO}}_{2}\times \mathrm{Qs})$. Once more, the arterial oxygen content (CaO_2_) will result from the sum of the two contents (10):

The content of the non-shunted fraction of the cardiac output (left term of the addition) and the content of the shunted fraction (right term of the addition)
(10)$${\mathrm{CaO}}_{2}={\mathrm{CcO}}_{2}\times \left(1-\frac{\mathrm{Qs}}{\mathrm{Qt}}\right)+\left({\mathrm{CvmixO}}_{2}\times \frac{\mathrm{Qs}}{\mathrm{Qt}}\right)$$

Based on the above equations, we can conclude that the native lung shunt fraction is the major determinant of the arterial oxygen content in partially dependent patients. Changes in cardiac output, oxygen consumption and haemoglobin concentration will affect the oxygen content in venous blood. A lower systemic venous oxygen content reduces the oxygen content in the pulmonary artery after mixing with the blood coming from the membrane lung, but increases the oxygen transfer in the ML, given that a lower venous saturation facilitates oxygen uptake (Figure 3). The relationship between ECBF and cardiac output is important as with maximal ECBF, decreasing the cardiac output (e.g., with the use of beta-blockers, lower temperature, sedation) can improve oxygenation in conditions when the cardiac output is high, and the oxygen extraction is impaired (e.g., fever or sepsis and relatively preserved high central venous oxygenation). However, clinicians need to be aware that a reduction in cardiac output can decrease total oxygen delivery and can therefore reduce venous oxygen content. The combination of reduced cardiac output (which should lead to increased PaO_2_) and decreased saturation of the venous blood (which should lead to decreased PaO_2_) can lead to a variable effect on the arterial oxygen content and partial pressure.

### 3.2. Carbon Dioxide Removal

#### 3.2.1. Determinants of CO_2_ Removal

Carbon dioxide transfer across the membrane lung is dependent on: (1) blood flow; (2) sweep gas flow; (3) membrane lung surface; (4) PCO_2_ gradient between the blood entering the membrane and the sweep gas flow [16,17]. We will discuss each individual element in the ECCO_2_R section, however it is worth pointing out here that, at the ECBF rates generally used in ECMO, CO_2_ removal—and therefore PaCO_2_—are dependent on the SGF rate. The SGF is effectively analogous to the minute ventilation in the native lung and its rate is in fact the only component used in clinical practice to adjust ECMO CO_2_ clearance.

ECMO can remove all of the metabolically produced CO_2_ and therefore maintain PaCO_2_ within normal range even in the absence of native lung ventilation (apnoea). The CO_2_ content of the venous blood is approximately 500 mL/L (400–450 mL/L in arterial blood), whilst the O_2_ content is 150 mL/L (200 mL/L in arterial blood). Therefore, in a perfectly efficient system the total CO_2_ production (VCO_2_) of approximately 250 mL/min could be removed by a blood flow of 0.5 L/min [16,18,19,20,21,22]. However, the O_2_ uptake of the same blood volume would only be 25 mL/min. This makes the respiratory quotient (VCO_2_/VO_2_) of the membrane lung highly variable and dependent upon the extracorporeal blood flow [23,24,25,26].

The total amount of CO_2_ removed will also depend on the function of the native lung. Therefore, the total amount of CO_2_ removed (VCO_2 TOT_) is:(11)$${\mathrm{VCO}}_{2\;\mathrm{TOT}}={\mathrm{VCO}}_{2\;\mathrm{ML}}+{\mathrm{VCO}}_{2\;\mathrm{NL}}$$

At equilibrium and for any given PaCO_2_, the total CO_2_ removed is the sum of ${\mathrm{VCO}}_{2\;\mathrm{ML}}+{\mathrm{VCO}}_{2\;\mathrm{NL}}$ and is equal to the metabolic CO_2_. According to Equation (12), an increase in VCO_2ML_ will allow a reduction of VCO_2 NL_ and alveolar ventilation with consequent reduction in the mechanical ventilation load.

#### 3.2.2. Carbon Dioxide Transfer across the Membrane

Two methods can be used to calculate the amount of CO_2_ removed by the membrane lung (VCO_2ML_):(1)The first method is based on the difference in CO_2_ content of the blood at each side of the membrane:
(12)$${\mathrm{VCO}}_{2\;\mathrm{ML}}=\;\left(C_{\mathrm{pre}-\mathrm{oxy}}{\mathrm{CO}}_{2}-C_{\mathrm{post}-\mathrm{oxy}}{\mathrm{CO}}_{2}\right)\;\times \mathrm{ECBF}\;\times 25$$

Equation (13): VCO_2ML_ (mL/min) as calculated from the trans-membrane CO_2_ content difference. C_post-oxy_CO_2_ is the CO_2_ content in the blood that exits the membrane lung (mmol/L), C_pre-oxy_CO_2_ is the CO_2_ content in blood entering the membrane (mmol/L). Blood flow is measured in L/min and the correction factor (25) is in mL/mmol.


(2)Alternatively, we can measure the concentration of CO_2_ from the sweep gas outlet of the membrane:
(13)$${\mathrm{VCO}}_{2\;\mathrm{ML}}={\mathrm{PCO}}_{2\;\mathrm{ML}\;\left(\exp\right)}\times \left(\frac{7.5}{713}\right)\times \mathrm{SGF}\times 1000$$


Equation (14): VCO_2ML_ (mL/min) is calculated from the partial pressure of CO_2_ in the effluent gas, PCO_2ML(exp)_ (kPa). The correction factors are 7.5 mm Hg/kPa and 713 mm Hg (barometric pressure at the sea level minus vapour pressure); SGF is measured in L/min.

In order to describe the VCO_2_ in standardised conditions, corrected VCO_2_ can be expressed as:(14)$${\mathrm{VCO}}_{2\;\mathrm{ML}\;\left(\mathrm{corr}\right)}={\mathrm{VCO}}_{2\;\left(D\right)}\times \frac{6}{{\mathrm{PCO}}_{2\;\mathrm{ML}\;\left(\mathrm{pre}\right)}}$$

Equation (15): VCO_2ML (corr)_ is corrected for an inlet PCO_2_ of 6 kPa using the Douglas equation. VCO_2 (corr)_ is measured in mL/min. PCO_2ML (pre)_ is the partial pressure of CO_2_ content in blood before the membrane (kPa).

#### 3.2.3. Membrane Dead Space

Membrane lungs are known to be affected by inequalities in the distribution of sweep gas and blood flows leading to local ventilation-perfusion mismatch [27]. Dead space in the membrane lung (i.e., proportion of the membrane receiving sweep gas flow but not in contact with blood flow) can be caused by water saturation or clotting of the gas capillaries. Analogously to the dead space of the NL, the dead space for the ML can be calculated as follows:(15)$${\mathrm{DS}}_{\mathrm{ML}}=\frac{P_{\mathrm{post}-\mathrm{oxy}}{\mathrm{CO}}_{2}-P_{\exp}{\mathrm{CO}}_{2}}{P_{\mathrm{post}-\mathrm{oxy}}{\mathrm{CO}}_{2}}$$

### 3.3. Monitoring of Gas Exchange

The following measurements are useful tools to track the progress of an individual patient and to assess the relative contribution of the membrane and native lung:A pre-oxygenator blood gasA post-oxygenator blood gasAn arterial blood gasA mixed venous blood gas (if a pulmonary artery catheter is in place)Expired PCO_2_ from the sweep gas port (if possible)Native lung minute ventilationVCO_2_ native lung (via volumetric capnography)

These should be assessed daily—from these values, it is then possible to calculate all the variables described in the ECMO physiology section.

### 3.4. ECMO Initiation: Physiological Considerations

Once a patient is placed on ECMO, the native lung function may deteriorate as a consequence of two main physiological phenomena:(1)Abolition of hypoxic vasoconstriction due to the increased oxygen in the mixed venous blood as a consequence of ECMO;(2)Changes in the respiratory quotient of the natural lung (that is consequent to the decarboxylated blood that arrives to the pulmonary artery);

The abolition of the hypoxic vasocontriction increases the physiological shunt of the native lung (Figure 4), while the reduction in the respiratory quotient (RQ) of the natural lung causes of a progressive fall in the alveolar PO_2_ based on the modified alveolar gas equation:(16)$${\mathrm{PAO}}_{2}=\left[{\mathrm{FiO}}_{2}-P_{\mathrm{atm}-H2O\;}\right]-\left(\frac{{\mathrm{PaCO}}_{2}\times {\mathrm{VO}}_{2}}{{\mathrm{VCO}}_{2\mathrm{NL}}}\right)$$

### 3.5. Weaning, “Trial off” and Decannulation

As the condition causing respiratory failure improves, the ECMO flow rate is reduced to a minimum of 2.5–3 L/min (to minimise the risk of thrombus formation at lower ECBF), while the SGF can be gradually reduced until it is turned off. In this phase, the patient is essentially “off ECMO” and can be decannulated if the gas exchange is stable and the ventilation is lung protective after 12–24 h. It is important to have a standardised weaning plan in place [28]. The pre-requisites for weaning assessment are haemodynamic stability and spontaneous or assisted ventilatory mode (e.g., cPAP/PS). Other additional parameters are: PaO_2_ > 30 kPa (225 mm Hg) after a 100% test (i.e., the systemic arterial PaO_2_ taken after 15 min on FiO_2_ = 1.0); P0.1 < 5 cmH_2_O.


Step 1: Decrease F_d_O_2_


FdO_2_ is decreased from 100% to 60%, 30% and 21% in 5–10 min steps. A peripheral oxygen saturation (SpO_2_) > 88% and P0.1 < 5 cm H_2_O will need to be maintained throughout the test. If this step is successful, FdO_2_ can be kept at 21%, and the test can proceed to step 2.


Step 2: Reduction in Sweep Gas Flow


The SGF is reduced by 30% every 5–10 min, whilst monitoring the patient’s respiratory efforts. Weaning failure and test interruption are indicated by SpO_2_ < 88%, RR > 35 bpm, P0.1 > 10 cm H_2_O, oesophageal pressure swings ≥15 cm H_2_O (when available), or if any signs of distress/instability are evident. If the patient’s response remains within set limits at 0 SGF, the weaning test is successful, and the clinical team will consider whether to remain off ECMO or reintroduce a variable degree of extracorporeal support pending decannulation.

### 3.6. Physiology of ECCO_2_R

ECCO_2_R, unlike ECMO, provides decarboxylation without oxygenation. CO_2_ removal is a more efficient process compared to oxygenation and can be achieved with lower blood flow rates (0.5–1.5 L/min) than the ones required for ECMO (3–7 L/min). For this reason, ECCO_2_R is sometimes referred to as “low-flow ECMO” [18,29]. The possibility of employing low flows allows the use of smaller size cannulae. However, at the same time, there is an increased thrombotic and haemolytic risk. ECCO_2_R has yet to find a definite place in standard care, with its use mainly limited to case series or within clinical trials.

The artificial membrane lung allows for dissociation between VCO_2_ and VO_2_. In the native lung, VO_2_ and VCO_2_ are physiologically strictly associated, with the metabolic quotient (R = VCO_2_/VO_2_) ranging between 0.7 and 1. In the presence of extracorporeal support, CO_2_ removal and O_2_ supply become independent given that R for the membrane lung may range between 0 and infinity, depending on the characteristics and composition of blood and gas flows. R = 0 is the equivalent of “apnoeic oxygenation of the membrane lung”, where there is no removal of CO_2_ in the presence of positive oxygen supply. Conversely, R = infinity is a consequence of extracorporeal CO_2_ removal in the absence of oxygenation via the membrane lung. This is achieved when the PO_2_ of the SGF and the PO_2_ of the venous blood perfusing it (inflow PO_2_) are equal. In these conditions, ECCO_2_R provides an R close to infinity, as it primarily removes CO_2_ while meaningful oxygenation is provided only by the native lung.

We will now focus on the physiological basis of carbon dioxide removal, exploring the ways in which it differs from the oxygenation process and the clinical implications that follow.

### 3.7. Principles of CO_2_ Diffusion and Transport

The total CO_2_ stores in the body are about 1.7–1.8 L/kg (e.g., 122–123 L of CO_2_ in a 70 kg individual), distributed within the lungs, blood, and other tissues, most of it present in the bones. The CO_2_ transported in the blood—in the form of bicarbonate ions, carbamino compounds, carbonic acid and a small amount of physical CO_2_—only accounts for 2.5–2.7 L. The vast majority of total body CO_2_ is contained in body tissues as bicarbonate, carbamino-CO_2_ and carbonate. This CO_2_ reserve is in continuous equilibrium with the blood component, which is the fraction of CO_2_ directly mobilised in the form of volatile gas by the natural lung and, when present, the membrane lung.

The use of low blood flow for ECCO_2_R is justified by physicochemical principles that make CO_2_ removal a much more efficient process than oxygenation. Carbon dioxide is more soluble and diffusible than oxygen. CO_2_ enters blood from the tissues and combines with water, forming carbonic acid (H_2_CO_3_), which in turn dissociates into hydrogen (H^+^) and bicarbonate ions (HCO_3_^−^). This reaction is catalyzed by carbonic anhydrase (CA), an enzyme present inside the red blood cell. As a result, most of the CO_2_ is transported in blood as bicarbonate:(17)$${\mathrm{CO}}_{2}+H_{2}O↔H_{2}{\mathrm{CO}}_{3}↔H^{+}+{\mathrm{HCO}}_{3}^{-}$$

The reverse reaction, that generates CO_2_ from HCO_3_^−^, follows linear kinetics and, unlike the bonding of oxygen to haemoglobin, does not become saturated [29]. Therefore, CO_2_ diffuses more efficiently than O_2_ and is not affected by haemoglobin concentration [30].

Haemoglobin plays an important role in the transport of CO_2_ as deoxyhaemoglobin increases the amount of carbon dioxide that is carried in venous blood by working as proton recipient.
(18)$${\mathrm{HbO}}_{2}+{\mathrm{CO}}_{2}+H_{2}O↔{\mathrm{HbH}}^{+}+{\mathrm{HCO}}_{3}^{-}+O_{2}$$

The binding of CO_2_ to Hb has relevant applications in increasing the efficiency of extracorporeal CO_2_ removal in low-flow systems.

### 3.8. Determinants of CO_2_ Transfer

The main determinants of CO_2_ transfer across the artificial membrane (VCO_2ML_) are:Extracorporeal blood flow (ECBF)Sweep gas flow (SGF)Membrane size and characteristicsCO_2_ gradient

#### 3.8.1. Extracorporeal Blood Flow (ECBF)

Typically, during high-flow veno-venous extracorporeal support, VCO_2ML_ increases linearly with gas flow and logarithmically with blood flow for a given inflow pCO_2_ [31]. In lower ECBF systems in the range of 0.1 to 0.5 L/min, an increase in ECBF will determine a linear increase in the CO_2_ removal (see also Equation (13)) [32]. As mentioned earlier, given that the CO_2_ content in 1L of venous blood is approximately 500 mL and the CO_2_ production is around 250 mL/min, an ECBF of 0.5 L/min would be theoretically sufficient to match the metabolic CO_2_ production. In reality, no system is perfectly efficient, and the currently available devices can achieve the removal of about 50–60% of the total CO_2_ production at best.

Despite the theoretical linearity of their relationship, the impact of blood flow on CO_2_ clearance is limited by the membrane size: if the surface area is not sufficient, increments in blood flow will not result in greater CO_2_ removal. The reverse is also true: low blood flow would not allow clinically significant CO_2_ removal regardless of the membrane size [33].

#### 3.8.2. Sweep Gas Flow (SGF)

SGF is one of the key determinants of CO_2_ transfer across the membrane lung: increments in SGF provide greater CO_2_ clearance. However, it is worth noting that SGF and VCO_2_ are not linearly related and this becomes especially apparent in low blood flow conditions: unlike ECMO, where SGF is the main parameter used to control CO_2_ removal, increasing the SGF in ECCO_2_R devices above a certain threshold (usually 4–5 L/min) does not result in further CO_2_ removal [34]. This ceiling effect is mainly determined by ECBF and membrane size.

#### 3.8.3. Membrane Characteristics

Membrane characteristics are another main determinant of CO_2_ transfer. The surface area has the greatest impact [33], but also membrane’s geometry and material affect efficiency of the process. Larger membranes are preferable in terms of CO_2_ clearance, given that a bigger surface is available for gas exchange. However, the larger the membrane, the greater its thrombotic risk. On the contrary, if the membrane is too small, there is higher blood velocity and increased potential for haemolysis.

In the experimental setting—where the measurement of VCO_2_ is more common than in clinical practice—the obtained VCO_2ML_ ranged between 35 and 75 mL/min for devices of 0.32 m^2^ [35] to 0.67 m^2^ [36], while it was 86 [37] to 170 mL/min when the lung surface was 1.8 m^2^ [32].

Membrane design can vary, too, with the sweep gas flow being either countercurrent or cross-current to the blood. Immobilized carbonic anhydrase on hollow fibers within membrane lungs can also be used to accelerate the conversion of carbonic acid into CO_2_ and water [38,39].

#### 3.8.4. Transmembrane pCO_2_ Gradient

The difference in pCO_2_ between blood and sweep gas is what drives the diffusion process, according to Fick’s diffusion law. Since CO_2_ concentration in the SGF is either very low or completely absent, the partial pressure of CO_2_ in the venous blood (PvCO_2_) captured by the inlet cannula of the ECCO_2_R (inflow PCO_2_) is what ultimately determines the diffusion gradient across the membrane lung. As CO_2_ diffuses and achieves equilibrium almost instantaneously, SGF rate is important to keep CO_2_ low on the gas side of the membrane.

For a given ECBF rate and membrane size, we can say that the VCO_2_ depends on the degree of hypercapnia—or transmembrane CO_2_ gradient. Indeed, a recent animal study on the use of an ECCO_2_R with membrane area of 1.8 m^2^ showed that the VCO_2_ was independent from SGF, while it increased linearly with ECBF. Inflow pCO_2_ (which drives the transmembrane PCO_2_ gradient) and ECBF were the primary determinants of VCO_2_, which was not affected by increasing gas flow above 4 L/min. This is most likely explained by the high efficiency of CO_2_ removal in the range of ECBF (100–400 mL/min) and inflow pCO_2_ (30–80 mm Hg) used in that study, as demonstrated by outflow pCO_2_ being close to its asymptote in all given conditions (approximately 1.3 kPa) [32].

### 3.9. CO_2_ Removal Rate

To maximize CO_2_ removal rate, research efforts are under way to artificially promote CO_2_ transfer by either driving the reaction (Equation (18)) towards increased volatile CO_2_ for techniques involving gaseous removal, or towards HCO_3_^−^ for approaches based on decarboxylation via a liquid medium.

#### 3.9.1. Gaseous CO_2_ Removal

The partial pressure of CO_2_ is only 5% of total CO_2_ content, corresponding to 12.5 mL of dissolved CO_2_ in 500 mL of blood. The conversion of bicarbonate and carbonic acid into CO_2_ is a slow reaction which is not sufficient to promote further CO_2_ removal [40]. An artificial increase in CO_2_ removal can be promoted by increasing FdO_2_ from 21% to 100% in the sweep gas, promoting the displacement of CO_2_ from HCO_3_^−^ through the Haldane effect [41].

Blood acidification represents another approach. Acidification of haemoglobin by oxygen binding has two main effects. Firstly, it facilitates CO_2_-Hb dissociation by reducing the binding affinity of haemoglobin for carbon dioxide, hence increasing the amount of CO_2_ available for gas exchange in the membrane. Secondly, it promotes the formation of carbon dioxide and water (Equation (18)), increasing again the amount of physical CO_2_ (PCO_2_) available.

The dilution of sulfur dioxide (SO_2_) in oxygen sweep gas acidifies blood, displacing the bicarbonate to CO_2,_ and allows 17% more CO_2_ removal while maintaining blood pH within physiological range [42].

Respiratory electrodialysis is a regional acidification technique which is able to achieve removal of 50% of total CO_2_ metabolism production with a blood flow of 250 mL/min, without systemic acidification. However, this strategy remains experimental and is not yet available on the market [43].

#### 3.9.2. Liquid CO_2_ Removal

Another strategy to remove CO_2_ is to drive the same reaction (Equation (18)) backwards by removing HCO_3_^−^ using a liquid alkaline dialysate with zero bicarbonate. Federspiel and colleagues were able to remove up to 117 mL/min of CO_2_ (inlet pCO_2_ 100 mm Hg and blood flow 421 mL/min) without affecting the systemic pH [44]. Later, using an innovative membrane with an integrated impeller in the oxygenator, they were able to remove 74 mL/min of CO_2_ with a blood flow of 250 mL/min and minimum haemolysis [41].

### 3.10. Potential Clinical Applications

Given its strong physiological rationale, ECCO_2_R undoubtedly represents an appealing technique, but one that still lacks sufficient evidence to support widespread clinical use [45]. Two main areas have been proposed for its clinical application: (1) acute respiratory distress syndrome (ARDS), where ECCO_2_R allows protective lung ventilation strategies; (2) acute type 2 respiratory failure, where hypercapnia represents the primary problem.


(1)Gattinoni and Kolobow first described the use of ECCO_2_R in ARDS in the late 1970s [23,24,26], following the affirmation of the “baby lung” concept and the development of lung protective ventilation strategies [46,47]. ECCO_2_R was suggested as an adjunctive tool that enabled ventilation with low tidal volumes (V_T_ 6–8 mL/Kg PBW) to prevent further lung injury, whilst addressing the hypercapnia resulting from a reduced minute ventilation. However, protective lung ventilation remains feasible in many situations without the use of ECCO_2_R, particularly when tolerating a moderate degree of hypercapnia (“permissive hypercapnia”). More recently, the role of ECCO_2_R in ARDS has gained renewed interest [48,49,50], often in parallel with the predicament of ultra-protective lung ventilation (V_T_ 3–4 mL/Kg PBW) [51,52,53]. Such approach does not provide adequate CO_2_ clearance and an alternative strategy for CO_2_ removal, e.g., extracorporeal support, becomes necessary.(2)Another potential application is in acute type 2 respiratory failure, such as acute exacerbations of obstructive airway disease (COPD and severe acute asthma). The postulated role for ECCO_2_R is to correct the acute element of respiratory acidosis, allowing the avoidance of intubation in patients with COPD exacerbation, or facilitating extubation in those patients already on invasive ventilation [54,55,56,57]. A theoretical benefit has also been suggested in severe acute asthma [58].


### 3.11. Effects of ECCO_2_R on Intensity of Mechanical Ventilation

The rationale for ECCO_2_R use in ARDS is to allow ultra-protective lung ventilation, with lower tidal volumes and lower driving pressures. These are both associated with improved survival. Goligher and co-authors describe how to predict changes in minute ventilation and driving pressure after applying ECCO_2_R [59]. The following equation is based on a theoretical analysis of the equation that describes the relationship between alveolar ventilation and CO_2_ clearance:(19)$$\Delta P=\frac{-\kappa }{C_{\mathrm{RS}}\left(1-\frac{V_{D\;\mathrm{alv}}}{V_{T}}\right)\cdot \mathrm{RR}\cdot {\mathrm{PaCO}}_{2}}\cdot {\mathrm{VCO}}_{2\;\mathrm{ML}}$$

Equation (20): DP is the predicted difference in driving pressure following initiation of ECCO_2_R. C_RS_, respiratory compliance. V_D alv/_V_T_, alveolar dead space fraction. PaCO_2_, CO_2_ partial pressure in arterial blood. VCO_2ML_, amount of CO_2_ cleared by membrane lung. k, correction factor (0.863).

According to this equation, the predicted change in driving pressure—at a given amount of extracorporeal CO_2_ removal—will be greater in those patients that have lower respiratory compliance and higher alveolar dead space fraction. To validate their hypothesis, they tested the prediction model utilising data from the SUPERNOVA trial [60], a pilot trial that aimed to assess efficacy and safety of ECCO_2_R in achieving ultra-protective lung ventilation in ARDS. The accuracy of the model was found to be only moderate, however the study confirmed that ECCO_2_R reduces the requirements for tidal volume and driving pressure in proportion to respiratory system compliance and alveolar dead space fraction [59]. The authors suggest important implications for future clinical trials: patients could be enrolled on the basis of their likelihood of obtaining benefit from the intervention—what is known as predictive enrichment. Namely, the enrollment criteria for ECCO_2_R in ARDS would include low respiratory compliance and high alveolar dead space fraction, as demonstrated above. This should increase the statistical power of trials and allow a smaller sample size. Ongoing research in this direction might help to define a specific clinical role for this extracorporeal modality.

## 4. Conclusions

The use of extracorporeal life support in respiratory failure is growing rapidly, both in numbers and indications. Understanding the physiology of native and membrane lung—the individual role they play as well as their interaction at different stages of disease and recovery—is key to successful use of ECLS in clinical practice.

## Figures and Tables

**Figure 1 membranes-11-00225-f001:**
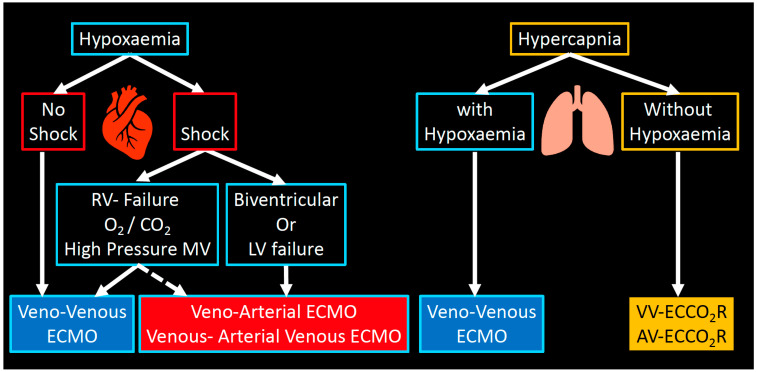
ECLS modalities. Overview of the available extracorporeal support modalities and their role in the management of respiratory failure.

**Figure 2 membranes-11-00225-f002:**
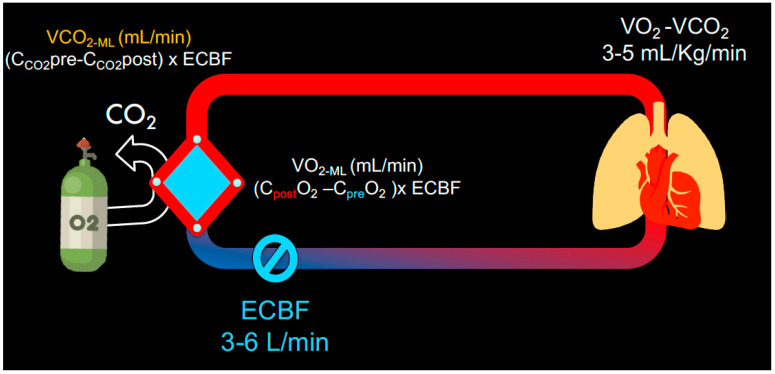
Schematic displays the oxygenation and decarboxylation of venous blood via the membrane lung. The VO_2_ and VCO_2_ are dependent on the content pre and post membrane and on the extracorporeal blood flow within the ECMO circuit.

**Figure 3 membranes-11-00225-f003:**
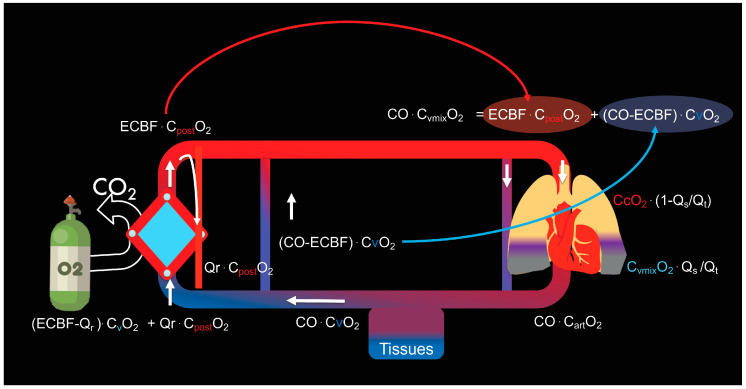
Schematic overview of the determinants of arterial PaO_2_. This is influenced by the degree of shunted deoxygenated blood from native lung, the ratio of ECBF to intrinsic cardiac output and amount of recirculation. Essentially, venous blood is divided into blood that is pumped through the ECMO membrane (ECBF) and an amount (Cardiac output minus ECBF) that bypasses the ECMO and mixes in the right atrium. Part of the ECBF (Qr) ‘recirculates’ back into the ECMO, decreasing the effective ECBF. Therefore, the effective ECBF (ECBFeff) is the difference between the ECBF and Qr. The oxygen content in the different compartments is illustrated.

**Figure 4 membranes-11-00225-f004:**
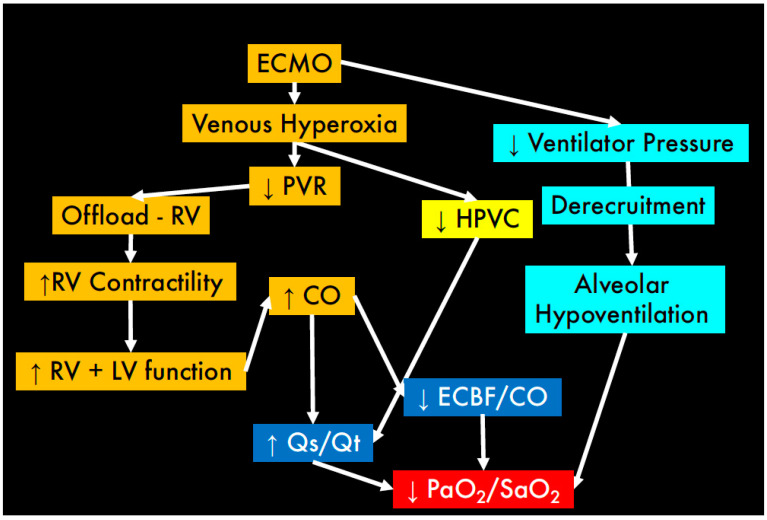
ECMO initiation. The delivery of VV-ECMO optimises oxygenation and allows a reduction of mechanical power delivered to the lungs. ECMO produces venous hyperoxia which in turn reduces pulmonary vascular resistance and hypoxic pulmonary vasoconstriction. This improves right ventricular function and cardiac output. A consequence of this physiological change is a larger shunt fraction in the native lung and a lower ECBF/CO ratio, both resulting in a lower arterial oxygen tension. This is further compounded by “lung rest” strategies leading to progressive de-recruitment and alveolar hypoventilation. In order to maintain adequate oxygenation ECBF may need to be increased to re-balance this ratio.

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
