# Peer review of "Physiological Basis of Extracorporeal Membrane Oxygenation and Extracorporeal Carbon Dioxide Removal in Respiratory Failure"

_membranes, 2021, doi:10.3390/membranes11030225_

Round 1
Reviewer 1 Report
Excellent contribution in the field of gas exchange in patients with ECLS.
I would ask the authors to expand the topic of CO/ECBF with particular details on the role of beta blockers.
Author Response
We thank the reviewer for her or his comment
Yes, we have added a short sentence as suggested:
Now we have added the following paragraph:
The relationship between ECBF and cardiac output is important as with maximal ECBF, decreasing the cardiac output (e.g., with the use of beta-blockers, lower temperature, sedation) can improve oxygenation in conditions when the cardiac output is high, and the oxygen extraction is impaired (e.g., fever or sepsis and relatively preserved high central venous oxygenation). However, clinicians need to be aware that a reduction in cardiac output can decrease total oxygen delivery and can therefore reduce venous oxygen content. The combination of reduced cardiac output (which should lead to increased PaO2) and decreased saturation of the venous blood (which should lead to decreased PaO2) can lead to a variable effect on the arterial oxygen content and partial pressure.
Reviewer 2 Report
In this manuscript the authors review the physiological aspects of ECMO and extra corporeal CO2 removal in patients with respiratory failure.
The paper is well written and sound. However, there is an abundance of formulae that are sometimes only briefly touched, lack explanation (formula 10) and are not consistent ( such as formulae 4 and 5). This is also the case for the numerous abbreviations. I recommend the authors to closely review the formulae in order to prevent too large steps. This would improve readability. The paper would also greatly improve from a list of abbreviations.
Fig 1: I suggest to insert an arrow between RV-failure and Venous-arterial venous ECMO. The number of centers that use veno-arterial ECMO or venous-arterial-venous ECMO in case of RV failure is increasing. This may be accompanied by a short paragraph 3 that mentions arterial ECMO. This may be very short and may state that this is beyond the scope of the review.
Minor:
ECMO initiation: please insert blank
CO2 stores: perhaps add instead of other tissues most of it in bones
Please have a look at the symbols around formula 20.
Please adjust format references eg 10, 24, 46 51. Ref 15 does not seem to make much sense
Author Response
In this manuscript the authors review the physiological aspects of ECMO and extra corporeal CO2 removal in patients with respiratory failure.
The paper is well written and sound.
We thank the reviewer for her or his comment
However, there is an abundance of formulae that are sometimes only briefly touched, lack explanation (formula 10) and are not consistent (such as formulae 4 and 5).
Many thanks for the suggestions. We have expanded the explanation for formula 10 (which is a derivation of the shunt equation) and slightly modified the formulae in 4 and 5.
This is also the case for the numerous abbreviations. I recommend the authors to closely review the formulae in order to prevent too large steps. This would improve readability. The paper would also greatly improve from a list of abbreviations.
Many thanks for the suggestions. We have included a comprehensive list of abbreviations and added some explanation to improve understanding of the formulae . We have modified also some figures and legends.
We have modeied and expanded Figure 3
Fig 1: I suggest to insert an arrow between RV-failure and Venous-arterial venous ECMO. The number of centers that use veno-arterial ECMO or venous-arterial-venous ECMO in case of RV failure is increasing. This may be accompanied by a short paragraph 3 that mentions arterial ECMO. This may be very short and may state that this is beyond the scope of the review.
Once again many thanks – we have done as suggested
We have added the following paragraph
veno-arterial extracorporeal membrane oxygenation (VA-ECMO). VA-ECMO provides support for the heart (e.g., biventricular failure). Hybrid configurations for peripheral ECMO (V-VA-ECMO), where there is drainage of the venous system and return into a central artery and a central vein to assist both heart and lung (e.g., severe ARDS and septic cardiomyopathy with cardiogenic shock).
Minor:
ECMO initiation: please insert blank
Done - Thanks
CO2 stores: perhaps add instead of other tissues most of it in bones
Done - Thanks
Please have a look at the symbols around formula 20.
Done – Thanks and adjusted
Please adjust format references eg 10, 24, 46 51. Ref 15 does not seem to make much sense
Done – Thanks and adjusted
Reviewer 3 Report
The present review reports on veno-venous ECMO and extra-corporeal carbon dioxid removal in patients with lung failure and the physiological basis of both methods. This review is extremely informative and interesting. It is very well written.
Please give a list of abbreviations used in the formulas before these abbreviations are used to make it easier for the reader to follow. At the moment it seems that there are no explanations abbreviations after first use nor an abbreviation list.
I got only minor issues to address.
Minor issues:
- To our point of view the term “ECLS” refers only to VA-ECMO and not to VV-ECMO or ECCO2R. Please eliminate to avoid confusion.
- Please give a list of abbreviations used in the formulas before these abbreviations are used to make it easier for the reader to follow.
- Abbreviations should be reduced.
- To increase clarity, please use the term “VV-ECMO” throughout the manuscript instead of “ECMO”
Author Response
The present review reports on veno-venous ECMO and extra-corporeal carbon dioxid removal in patients with lung failure and the physiological basis of both methods. This review is extremely informative and interesting. It is very well written.
We thank the reviewer for her or his comment
Please give a list of abbreviations used in the formulas before these abbreviations are used to make it easier for the reader to follow. At the moment it seems that there are no explanations abbreviations after first use nor an abbreviation list.
Done – Thanks and adjusted
I got only minor issues to address.
Minor issues:
- To our point of view the term “ECLS” refers only to VA-ECMO and not to VV-ECMO or ECCO2R. Please eliminate to avoid confusion.
The Maastrich treaty for nomenclature in ECMO now considers VV-ECMO, VA ECMO and ECCO2R as part of ECLS (see link). So, we have chosen to maintain the nomenclature that is concordant with the Maastricht Treaty
https://www.atsjournals.org/doi/pdf/10.1164/rccm.201710-2130CP
- Please give a list of abbreviations used in the formulas before these abbreviations are used to make it easier for the reader to follow.
Done – Thanks and adjusted
- Abbreviations should be reduced.
We have done so as much as possible – with thanks
- To increase clarity, please use the term “VV-ECMO” throughout the manuscript instead of “ECMO”
Many thanks we have added a sentence
From this point forward when we mention ECMO we intend the veno-venous (VV) configuration.